# Highly Mesoporous g-C_3_N_4_ with Uniform Pore Size Distribution via the Template-Free Method to Enhanced Solar-Driven Tetracycline Degradation

**DOI:** 10.3390/nano11082041

**Published:** 2021-08-11

**Authors:** Bao Lee Phoon, Chin Wei Lai, Guan-Ting Pan, Thomas C.-K. Yang, Joon Ching Juan

**Affiliations:** 1Nanotechnology & Catalysis Research Centre (NANOCAT), Level 3 Block A, IPS Building, Institute for Advanced Studies, University of Malaya, Kuala Lumpur 50603, Malaysia; phoonpauline@um.edu.my (B.L.P.); cwlai@um.edu.my (C.W.L.); 2Department of Chemical Engineering, National Taipei University of Technology, Taipei City 10608, Taiwan; t6679013@gmail.com (G.-T.P.); ckyang@mail.ntut.edu.tw (T.C.-K.Y.); 3School of Science, Sunway Campus, Monash University, Jalan Lagoon Selatan 47500, Malaysia

**Keywords:** tetracycline, mesoporous, graphitic carbon nitride, pore size, photocatalyst, g-C_3_N_4_

## Abstract

A highly mesoporous graphitic carbon nitride g-C_3_N_4_ (GCN) has been produced by a template-free method and effectively photodegrade tetracycline (TC) antibiotic under solar light irradiation. The mesoporous GCN (GCN-500) greatly improves the photoactivity (0.0247 min^−1^) by 2.13 times, as compared to that of bulk GCN (0.0116 min^−1^). The efficiently strengthened photoactivity is ascribed to the high porosity (117.05 m^2^/g), and improves the optical absorption under visible light (E_g_ = 2.65 eV) and good charge carrier separation efficiency. The synthesized mesoporous GCN shows a uniform pore size (~3 nm) distribution. GCN-500 shows large pore volume (0.210 cm^3^/g) compared to GCN-B (0.083 cm^3^/g). Besides, the GCN-500 also exhibits good recyclability and photostability for TC photodegradation. In conclusion, GCN-500 is a recyclable photocatalyst for the removal of TC under visible light irradiation.

## 1. Introduction

Antibiotics are a medicine commonly used for human treatment and by veterinarians for bacterial disease treatment [1]. Nonetheless, antibiotics have often been found in the surrounding environment because of the incomplete metabolism from poultry farms and humans. The trace amount of antibiotics in water can be linked to human health hazards [2] and serious environmental problems [3]. In recent years, Asghar and co-workers found that the Pearl River in China was contaminated with various types of antibiotics at concentrations of 1.39 µg/L [4]. There are miscellaneous antibiotic products available for different purposes. Among these, tetracycline (TC) is one of the highest use antibiotics in the world [5]. Unfortunately, up to 80% of the incomplete TC metabolites could be excreted or released into domestic waste. The uptake of this TC was found to cause various diseases, such as allergies in humans [6].

The removal of TC has been widely studied by using different kinds of processes, such as photocatalysis [7], adsorption [8], the biological method [9], and membrane process [10]. Adsorption can generate secondary waste, which is less practical in the industry [11], and the cost of regeneration of adsorbent is relatively high. The membrane process can possibly cause the fouling effect [12], in which the membrane needs to be replaced occasionally. The biological method requires environment optimization [13], and the treatment process is relatively slow. Photocatalysis is a promising technology because it can break down organic pollutants such as antibiotics into CO_2_ and water by using natural solar light [14]. Since solar consists of up to 50% of visible light, the photocatalyst should also inherit a small bandgap to be activated in visible light. There are many visible light-actived photocatalysts, such as N-doped TiO_2_, Ag_3_PO_4_, and CdS. However, the visible light absorption of N-doped TiO_2_ is only limited to the color purple with a bandgap of ~2.95 eV [15]. Meanwhile, the bandgap of Ag_3_PO_4_ and CdS is small, but the photostability is not good [16,17]. Recently, carbonaceous metal-free photocatalysts attracted great interest because it is consisting of earth-abundant non-metallic elements, environmental friendliness, and low cost [18].

There are several carbon-based metal-free photocatalysts, such as carbon nanotubes and graphene, that have been widely reported by researchers. Among that, g-C_3_N_4_ (GCN) has unique characteristics due to visible light active with a bandgap of 2.4–2.9 eV and strong photochemical stability. The application of GCN as a photocatalyst has been studied as water splitting and photodegradation of organic pollutants. Nonetheless, the photoactivity of bulk GCN is limited due to its small surface area and non-porous nature [19]. Some previous study has been attempted to enlarge the surface area of GCN via different templates, namely, the silica hard template [20] and soft template technique [21]. These techniques require us to use harsh and corrosive chemicals such as hydrofluoric acid to remove the template from GCN after the synthesis process [22]. Aligned with the green chemistry principle and sustainable development goal 2030 (SDG 2030) [23], high surface area mesoporous GCN is synthesized via the template-free technique.

Several precursors have been used to produce mesoporous GCN, for example, urea-derived GCN for NO_x_ removal [24], dicyandiamide produced GCN for water splitting [25], and cyanamide synthesized GCN for dye removal [26]. It was reported by Guan and co-workers, type of precursors is one of the factors to influence the GCN photoactivity and melamine synthesized GCN was the highest performance for their study [27]. Guan and co-workers also reported that different precursors have different pore size distribution in their study [27]. For example, melamine and dicyandiamide precursors produced uniform pore size distribution, while urea precursor produced wide pore size distribution of GCN. Several researchers have done mesoporous GCN studies, but the pore size distribution was varied. For example, Sun and co-workers synthesized mesopore GCN with a pore size distribution in the range of 10–100 nm [24], Ge et al. synthesized mesoporous GCN for nitrogen photo-fixation application with pore size in the range of 20–160 nm [28], Hernandez-Uresti et al. produced mesoporous GCN for degradation with a pore size range of 5–100 nm, and these are wide pore size distributions [29]. Meanwhile, a mesoporous GCN photocatalyst with a uniform pore size distribution is rarely reported. It is known that the material pore structure is important, because the pores are able to reflect and scatter light to enhance the light response of the photocatalyst [26].

Hence, this study reports producing mesoporous GCN by a simple re-calcination method by using melamine as a precursor. The mesoporous GCNs was calcined in between 450 and 550 °C. The mesoporous GCNs exhibited high mesoporous (117.05 m^2^/g), large pore volume (0.210 cm^3^/g) uniform pore size distribution (3 nm), and slow charge recombination. Furthermore, they exhibit excellent photoactivity for TC removal.

## 2. Experimental

### 2.1. Preparation of Bulk GCN

To produce bulk GCN, 3 g of melamine powders (3 g, Sigma Aldrich, Germany, >99%) were put into an alumina crucible and covered with a lid. It was calcined at 550 °C in air for 4 h at a 3.0 °C min^−1^ heating rate as shown in Figure 1. This GCN was collected and ground into fine powders. The resultant GCN was named GCN-B.

### 2.2. Preparation of Mesoporous GCN

The mesoporous GCN samples were formed by recalcination. 3 g of GCN-B was transferred to the alumina crucible and covered with a lid. The crucible was annealed at three different temperatures (450, 500, and 550 °C) for 2 h at 3.0 °C min^−1^ to obtain mesoporous GCN, as illustrated in Figure 2. The resultant mesoporous GCNs were denoted as GCN-450, GCN-500, and GCN-550, respectively.

### 2.3. Characterization

A Bruker D8 diffractometer with Cu-Kα radiation (λ = 0.154 nm) was applied to measure powder X-ray diffraction (XRD). The applied current was 30 mA and the accelerating voltage was 40 kV. The range of data collection for XRD was 10–80° with 0.02 s^−1^ of the scan rate. The GCN samples’ structural information was determined by a Fourier transform infrared (FTIR, Perkin-Elmer 100 spectrophotometer) that was done at room temperature. The range of scans was 4000 to 400 cm^−1^ with a scan rate of 4 cm^−1^ with 8 scans. A Hitachi SU8230 field emission scanning electron microscopy (FESEM) that operating at 5 kV and a transmission electron microscope (TEM, FEI-Tecnai F20) that operating at 200 kV was applied to study the morphology GCNs. A Thermo A scientific K-alpha instrument using Mg Kα X-ray sources was applied to obtain the X-ray photoelectron spectroscopy (XPS) of GCN samples. C1s signal at 248.8 eV was used to correct binding energy shifts. A Tristar II 3020 porosity and surface area analyzer (Micromeritics Instrument Corporation, USA) was applied to obtain the nitrogen adsorption-desorption isotherms at 77 K. Before each measurement, the GCN samples were degassed for 4 h at 150 °C. Then, the average pore size determination was based on the Barrett–Joyner–Halenda (BJH) theory. A UV–Vis spectrophotometer (Perkin–Elmer Lambda 35, USA) was used to record the UV–Vis spectra with an integrating sphere attachment within the range of 200–800 nm. The optical bandgap of GCN samples was determined based on the Kubelka–Munk (K-M) function. The photoluminescence (PL) spectra were investigated with a PTI Model Quantamaster-QM4m spectrofluorometer equipped with dual excitation monochromators (325 nm) and 75 W lamp equipment.

### 2.4. Photodegradation Study

The GCNs were investigated via photodegradation of TC in aqueous with a 150 W Xe lamp irradiation (λ > 420 nm) to find out its photocatalytic activities. First, a 0.1 L of aqueous TC (10 mg/L) was mixed with 30 mg of GCN samples. Room temperature was maintained during the photocatalytic experiment process. The suspension was collected (4 mL) to clear the GCN photocatalyst by filtration for every 30 min interval. The reactor was kept in the dark for 90 min to achieve adsorption equilibrium. The TC was then been measured by its absorption peak intensity with a UV–Vis spectrophotometer to determine its percentage removal. Isopropyl alcohol (IPA), ethylenediaminetetraacetic acid disodium salt (EDTA), and benzoquinone (BQ) were employed as the scavengers for hydroxyl radical, holes, and superoxide radical, respectively to determine the key active species of TC degradation with 0.01 M concentration of all sacrificial agents.

### 2.5. Photoelectrochemical Study

An electrochemical analyzer, (µAutolab III) coupled with a 150 W Xe lamp (λ > 420 nm) was applied to study the photoelectrochemical properties of GCN. An Ag/AgCl electrode and Pt wire were employed as the reference and auxiliary electrode, respectively. The working electrodes were fabricated by using the electrophoretic deposition method were GCN samples and Mg(NO_3_)_2_•6H_2_O (purity and brand) dispersed in IPA to form a colloidal suspension. The distance between each electrode was 10 mm. The deposited working electrodes were rinsed with distilled water and transferred to an oven with 80 °C (3 h) for the drying process. Meanwhile, the impedance data was collected in a frequency range from 1 Hz to 100 kHz under 1 V vs. reference condition with 100 mV amplitude.

## 3. Result and Discussion

### 3.1. Physicochemical Properties of GCNs

Figure 3a presents the XRD diffractogram of all GCN samples is well-matched with GCNs (JCPDS No. 010871526). The peak at 13.09° corresponding to (100) plane and attributed to in-plane d-spacing. The peak at 27.60° represents the (002) plane, which is the interlayer stacking reflection of conjugated aromatic segments. Figure 3b shows the recalcined GCNs have a slight shift of 2θ to higher values, indicating that the interlayer spacing of recalcined GCNs slightly decreases [30]. The d-spacing values of the peak (002) and (100) are shown in Table 1. The (002) shows a significant decrease of d-spacing value while (100) shows an insignificant change. It was proven that interlayer distance and GCN are slightly decreased after calcination.

FTIR spectra in Figure 4 is used to determine the chemical bonding of GCN-B and GCNs at different calcination temperatures. The peaks at 807 and 889 cm^−1^ demonstrate that all the GCN structures are tri-s-triazine. Meanwhile, the peaks located in between 1243 and 1633 cm^−1^ reflected the stretching vibration of C-N heterocyclic, comprising both secondary and tertiary amine structures. This indicates that the extended network of C–N–C bonds has been formed [31]. The broadband in between 3000–3400 cm^−1^ represents the NH and NH_2_ stretching.

Figure 5 displays the morphology of GCN samples. The GCN-B shows disordered bulks stacking together. Obviously, GCN-B has a typical non-porous architecture and slightly flat. Figure 5b shows a flake-like surface of GCN-450, some of the holes were exposed on the surface. Meanwhile, refer to Figure 5c, the surface of GCN-500 is distributed with numerous inhomogeneous holes and uneven surfaces. This finding was concordance with Papailias et al. [32] because when the temperature of recalcination rises to 550 °C, the GCN structure shows loose surface morphologies. Furthermore, this experiment did not report the 600 °C of GCN recalcination because it was found that the GCN was fully disappeared after recalcination.

The nanostructure of GCN samples was further demonstrated by the TEM image. The GCN-B (Figure 6a) shows numerous bulks and the structures are stacked together, which is consistent with the observation by FESEM. Then, when the recalcination takes place at 450 °C, the layer of GCN-450 becomes thin compared to the GCN-B but some of the layers still overlapped. Meanwhile, GCN-500 shows that during the recalcination, the large layers are crack into tiny nanosheets. Then, when the recalcination temperature increase to 550 °C, those small sheets of GCN are dispersed evenly. As a result, the GCN sheets become thin and dispersed after recalcination, which is because of mass loss during recalcination. In addition, Figure 6e shows the lattice spacing of 0.319 nm between the (0 0 2) plane of GCN-500, the *d*-spacing value is well agreed with the XRD result.

Figure 7 shows the isotherms of N_2_ adsorption–desorption for GCN-B, and GCNs that recalcined at different temperatures. All the GCN show a similar isotherm and are classified as classical type V. It is recommended that all the surfaces of GCNs are mesoporous. Meanwhile, the hysteresis loop belongs to H3 which shows a parallel plate slit pore structure. Initially, the GCN-B specific surface area (S_BET_) was 30.20 m^2^/g. After recalcined at 450 °C, the S_BET_ of GCN increased to 79.96 m^2^/g. With the increase of recalcination temperature, the GCN surface area was also increased. The GCN-550 shows that the highest S_BET_, which was 158.57 m^2^/g. This indicates that after high-temperature recalcination, numerous pores are leaving on the GCN. Furthermore, the average pore size of GCN samples was listed in Table 1. All the pore diameters of GCNs are approximately 3 nm, which is mesopores. By increasing the recalcination temperature, the pore volume also increased. The highest pore volume was 0.228 cm^3^/g (GCN-550) and the lowest was 0.083 cm^3^/g (GCN-B). According to Liu et al., the amount of gas released during the GCN recalcination can affect the formation of the pores [33]. The gaseous released is due to the GCN slowly oxidize at high temperatures. In this case, the GCNs were recalcined in three different temperatures, therefore, it can be deduced that if the recalcination temperature is high, the GCN becomes highly porous due to the massive of gas released.

Table 1 shows the mass of GCNs are decreased after recalcination with the temperature increased. respectively. Meanwhile, GCN-550 is only left 18% of yield after recalcination. Appendix A shows the TGA of GCN-B. It has further been proven that the GCN-B can be decomposed at high temperatures. The volume changed reveals that the mesoporous GCNs possess a loose framework compared to GCN-B. As mentioned by Dong and co-workers, the changes in microstructure GCNs with recalcination are like the activation of charcoal [34]. A portion of the GCN might break down after the recalcination process, thereby leaving many pores that lead to the structure resulting GCNs porous. The elemental composition of GCN samples is tabulated in Table 1. It was found that the C/N ratio increased with recalcination temperature. The highest C/N ratio (0.809) was GCN-550, while the lowest C/N ratio was GCN-B, which is 0.681. This indicates that, after the high-temperature recalcination, the carbon content has increased, such as GCN-550. If the carbon content is high, it can affect the textural properties of CN for example high porosity of the surface [35].

Figure 8 shows the GCN-B and GCN-500 XPS spectra. According to the GCN spectra, there are two major peaks, that are placed at 287 eV for C1s and 398 eV for N1s, respectively. The GCN-B sample consist of three C1s peaks, namely 284.5 eV (carbon double bond) [36], 287.3 eV (C–(N)_3_) [37], and 288.4 eV (C–O bond) [38]. Nevertheless, the peak at 288.4 eV disappeared after recalcination, which indicates the C–O species has been oxidized. Figure 8b demonstrates that GCN-B is deconvoluted into three peaks in N1s spectra, which are 398.2 eV, 400.3 eV, and 401.7 eV that represents C=N–C, N–(C)_3_, and C–N–H amino functional groups from the structure of tri-s-triazine [39]. At the same time, the peak at 401.7 eV is not detected in GCN-500 (Figure 8d) because the C–N–H has been oxidized. This is also in concordance with the research work of others [40], in which the peak of 401.7 eV disappears after the GCN recalcined at high temperature. Appendix A shows the N1s peak of sample GCN-450 and GCN-550. The peak of 401 eV does exist for GCN-450 sample, but disappear in GCN-550. Besides that, the recalcination process of GCNs does not affect the O1s peak that presented in Appendix A.

### 3.2. Optical Properties of GCNs

Figure 9 presents the light absorption characteristics of GCN photocatalysts. According to Figure 9a, the GCNs have a slight hypsochromic shift after recalcination at 450, 500, and 550 °C. This is because of the quantum confinement effect [32], where light absorption of mesoporous GCN is slightly shifted to high energy region. The bandgap energy of GCN-550 has increased by 0.1 eV or ~4% only after recalcined at 550 °C as compared to GCN-B.

The optical characteristics of GCN-B and GCN samples that were treated at different temperatures have been accessed by PL (Figure 9c). The GCN samples emission peak was at 530 nm, as ascribed to the band-to-band PL phenomenon. Those recalcined GCN show a lower PL intensity than that of GCN-B. This indicates that the rate of recombination has been suppressed from the recalcination process. The recombination rate sequences are as follows: GCN-B > GCN-450 > GCN-550 > GCN-500. It was reported by Du and coworkers that the high carbon content with the mass fraction of GCN can improve its charge transfer [41]. As mentioned in Table 1, the carbon content of GCN was higher than GCN-B after recalcination at elevated temperatures. For example, the C/N ratio of GCN-500 was found to be 0.720 which ~6% higher than GCN-B. Thus, this finding is in good agreement with Wang et al., as the high C/N ratio of GCN greatly lowers the rate of recombination [21].

### 3.3. Photodegradation Performance of TC

The adsorption capacity gradually increases with the recalcination temperature. Figure 10a shows the photodegradation performance of 10 mg/L TC in visible light irradiation. It was found that recalcined GCN has better dark adsorption than GCN-B, and the adsorption capacity was gradually increased with the increase of recalcination temperature. It was noticed that the photodegradation efficiency sequences as follows GCN-500 (0.0247 min^−1^, 99.4%) > GCN-550 (0.0158 min^−1^, 96.0%) > GCN-450 (0.0137 min^−1^, 93.5%) > GCN-B (0.0116 min^−1^, 87.8%). The GCN-500 shows the highest photodegradation activity because of the large specific surface area (117.05 m^2^/g). A large specific surface area indicates that more active sites on GCN-500, that able to receive light irradiation and enhanced that photocatalysis process, confirming that 500 °C is the optimal recalcination temperature. In addition, GCN-500 has a visible light bandgap, which is 2.65 eV, corresponding to 468 nm. With these two criteria, visible light band gap and large surface area and pore volume, the photoactivity of GCN-500 to degrade TC has greatly improved. This is because GCN-500 is highly mesoporous, which is rich in active sites. It was found that the mesoporous property is also good in organic adsorption [42] and this helps to promote better photocatalytic performance. This is because once the pollutant adsorbs on the photocatalyst surface, photocatalysis can only take place. Besides that, a large pore volume able to capture more TC pollutants to the surface of the photocatalyst [43] and causing the degradation to become faster. Meanwhile, under the presence of visible light irradiation, GCN-B exhibits the lowest photodegradation efficiency of 0.0116 min^−1^ or 87.8% removal due to the low specific surface area (30.20 m^2^/g). Then, the photodegradation outcome was calculated with two models, which are pseudo-first order and pseudo-second-order. All the TC photodegradation kinetics were well fitted using the pseudo-first-order model with the correlation coefficient 0.95–0.98 (Appendix A). As compared with that GCN-B (k = 1.16 × 10^−2^ min^−1^), the photoactivity of GCN-500 was improved by about 2.13 times (k = 2.47 × 10^−2^ min^−1^).

The photocatalytic mechanism was evaluated with several radical scavengers, such as IPA (hydroxide radical), BQ (superoxide radical), and EDTA (holes). Based on Figure 10c, BQ has apparent changes as compared with the absence of scavenger. Meanwhile, there were no noticeable changes in the TC photodegradation when IPA is added. This indicates that hydroxide radical species is not the key species in TC photodegradation. According to these outcomes, the predominant active species were superoxide anions and holes. Thus, it can be proposed that when the mesoporous GCN was excited by visible light, holes and electrons were generated (Equation (1)). After that, the light-induced electron in the conduction band of mesoporous GCN reacted with oxygen on the catalyst surface to produce superoxide anions as in Equation (2). These superoxide anions are possible to further react with holes to produce hydroxyl radical. However, the hydroxyl radical was the least active species in TC photodegradation, because the potential to generate the hydroxyl radical was +1.99 eV (•OH/OH^−^), which is more negative than the valence band of GCN (+1.56 eV) [44]. In the meantime, these active species radicals could be attracted by TC and undergo chemical structure breakdown and produce smaller degraded products (Equations (4)–(6)).
(1)$$\mathrm{GCN}\overset{\mathrm{hv}}{\to }\;\mathrm{GCN}\left(h^{+}+e^{-}\right)$$
(2)$$e^{-}+O_{2}\to \;\cdot O_{2}^{-}$$
(3)$$2H^{+}+\cdot O_{2}^{-}\to 2\cdot \mathrm{OH}$$
(4)$$\mathrm{TC}+\cdot O_{2}^{-}\;\to \mathrm{degraded}\;\mathrm{products}$$
(5)TC+·OH →degraded products
(6)$$\mathrm{TC}+h^{+}\to \mathrm{degraded}\;\mathrm{products}$$

The GCN-500 was reused six times for photodegradation of TC. Figure 8d demonstrates that the photodegradation rate of GCN was mildly decreased to about 6% after six cycles, which indicated that GCN-500 is relatively stable.

Figure 10e presents the COD removal of TC by using GCN that recalcined at different temperatures in 30 min intervals. The purpose of COD is to determine the oxygen equivalent of the organic content in a sample that is susceptible to oxidation to water and carbon dioxide by a strong oxidant. The COD removal percentage of each of the samples was increased from time to time. This indicates that GCN samples can remove COD effectively in the pollutants. It was noticed that GCN-500 shows the highest COD removal after 180 min of photodegradation, which is 79.4%. However, the lowest COD removal was 64.9%, which is contributed by GCN-B. This can be explained by the fact that GCN-B was the slowest to be mineralized or break down the TC intermediates while the GCN-500 sample was the fastest to break down TC intermediates.

### 3.4. Photoelectrochemical Properties of GCNs

An impedance study was used to test the charge transfer performance of GCNs, as shown in Figure 11a. The prepared GCN-500 has the smallest semicircle among all the samples, which reveals the highly effective separation and transfer of the light-induced carrier in the GCN-500 sample. This concordance with PL that the GCN-500 has the slowest recombination rate. Furthermore, Figure 11b represents the responses of photocurrent for GCN-B and GCNs that recalcined at different temperatures with three light-on and light-off cycles under the irradiation of visible light. Certainly, the photocurrent of GCN-500 is 2.67 times greater than GCN-B, which allows for the more effective separation of charge carriers.

## 4. Conclusions

A highly-mesoporous GCN is synthesized using a template-free method with uniform pore size distribution. The GCN-500 displayed superior TC photodegradation activity up to 99.4% removal within 180 min. The enhanced photodegradation performance could be ascribed to the optimum bandgap (2.65 eV), high surface area (117.05 m^2^/g), large pore volume (0.210 cm^3^/g) and great charge carrier separation efficiency. The photogenerated h^+^ and O_2_^−•^ radicals as two main species that are significant in TC photodegradation through GCN-500. The GCN-500 is highly reusable as after recycling for six cycles, and the photodegradation performance mildly decreased by 6%.

## Figures and Tables

**Figure 1 nanomaterials-11-02041-f001:**
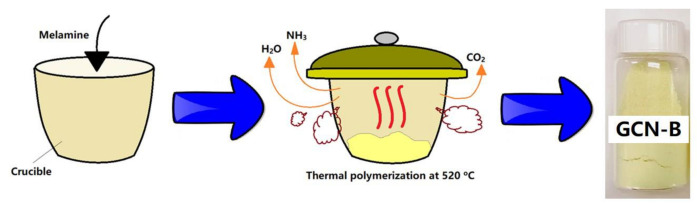
Schematic flow of the GCN synthesis process.

**Figure 2 nanomaterials-11-02041-f002:**
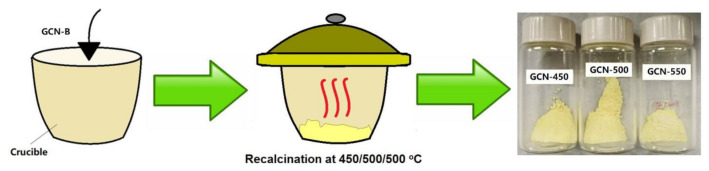
Schematic flow of the highly mesoporous GCN process.

**Figure 3 nanomaterials-11-02041-f003:**
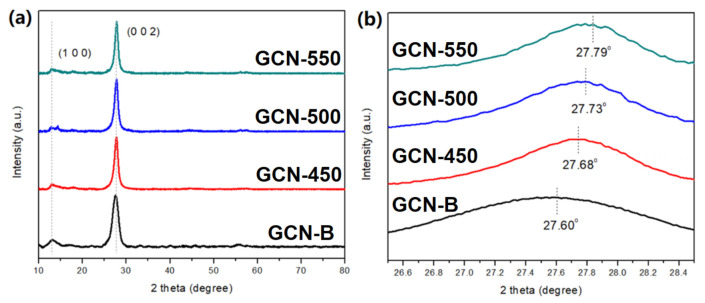
(**a**) XRD diffractogram of GCN-B and GCN that recalcined at different temperatures; (**b**) the enlarged peak of (002).

**Figure 4 nanomaterials-11-02041-f004:**
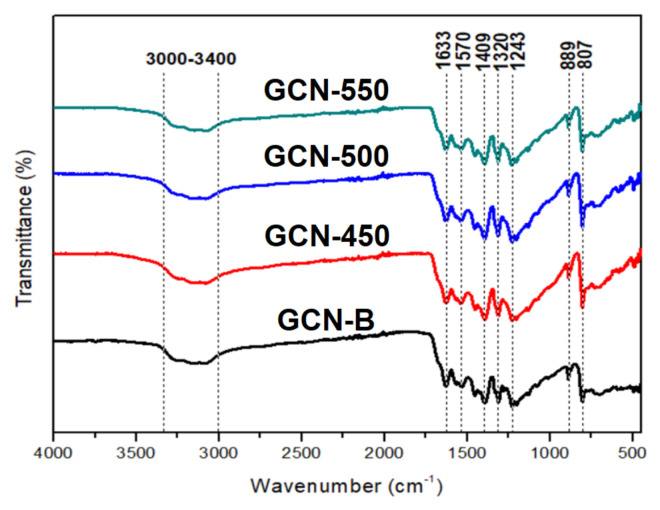
FTIR spectra of GCN-B and GCNs that recalcined at different temperatures.

**Figure 5 nanomaterials-11-02041-f005:**
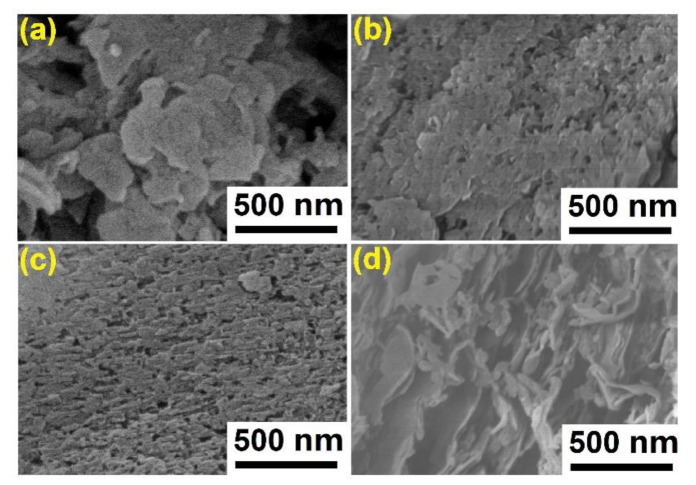
FESEM micrograph of GCN samples (**a**) GCN-B; (**b**) GCN-450; (**c**) GCN-500; (**d**) GCN-550.

**Figure 6 nanomaterials-11-02041-f006:**
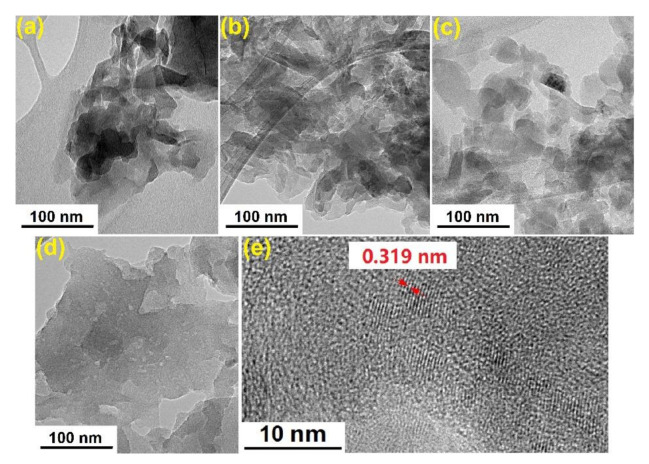
TEM images of (**a**) GCN-B; (**b**) GCN-450; (**c**) GCN-500; (**d**) GCN-550; (**e**) lattice spacing of GCN-500.

**Figure 7 nanomaterials-11-02041-f007:**
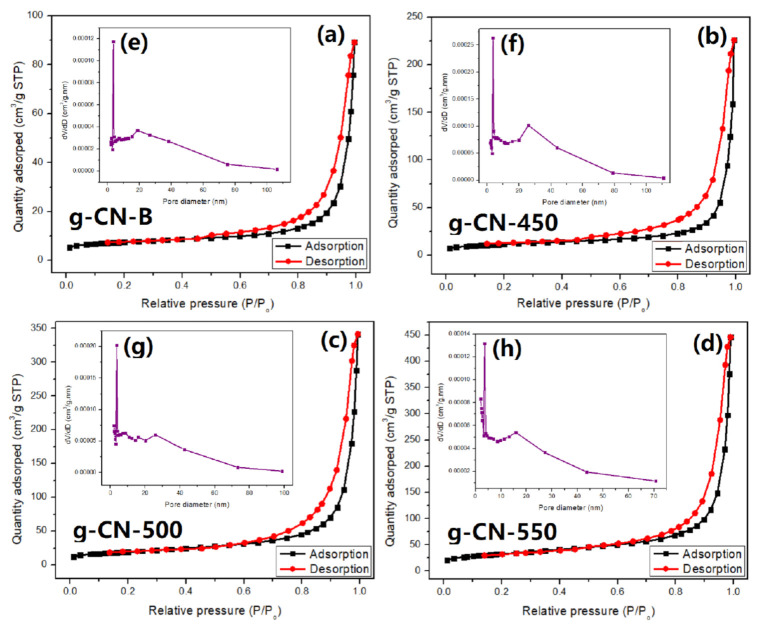
The N2 adsorption-desorption isotherms (**a**) GCN-B, (**b**) GCN-450, (**c**) GCN-500, (**d**) GCN-550, and the BJH pore size distributions of (**e**) GCN-B, (**f**) GCN-450, (**g**) GCN-500, (**h**) GCN-550.

**Figure 8 nanomaterials-11-02041-f008:**
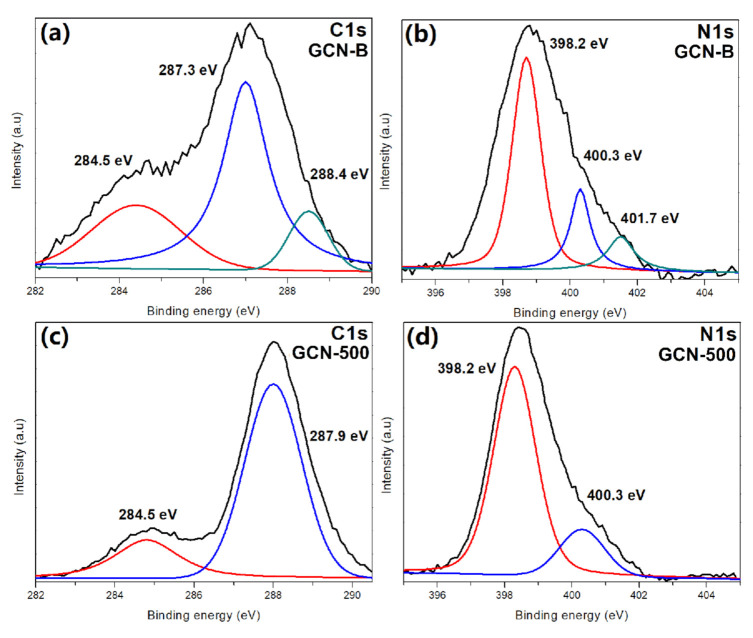
XPS spectra of GCN samples (**a**) C1s of GCN-B, (**b**) N1s of GCN-B, (**c**) C1s of GCN-500, (**d**) N1s of GCN-500.

**Figure 9 nanomaterials-11-02041-f009:**
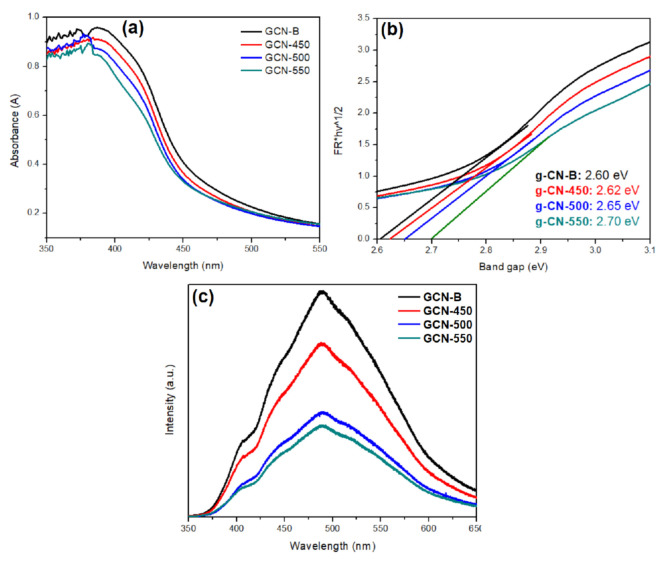
(**a**) UV–Vis absorption spectrum; (**b**) plots of (αhv)1/2 vs. photon energy of the GCN-B and GCN that recalcined at different temperature; (**c**) PL spectra of GCN-B and GCN that recalcined at different temperature.

**Figure 10 nanomaterials-11-02041-f010:**
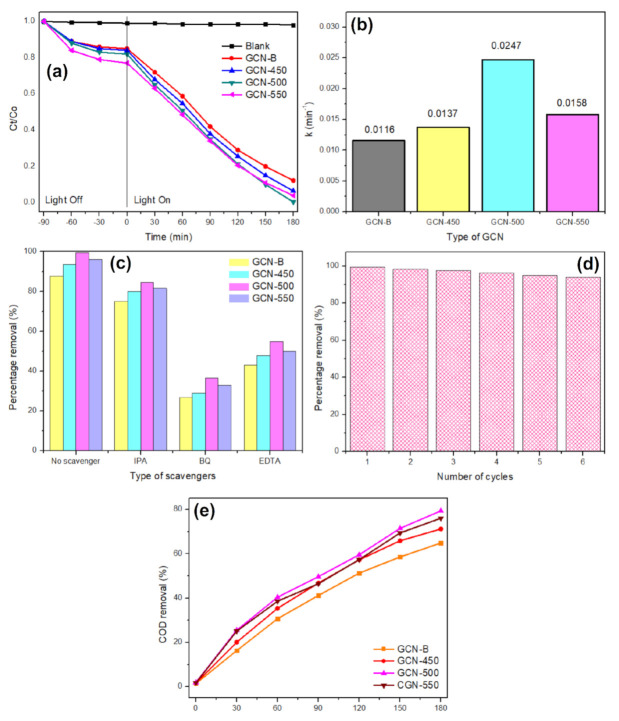
(**a**) The photocatalytic performance of TC with GCN samples under irradiation of visible light (λ > 420 nm), (**b**) the kinetics for TC photodegradation with different GCN samples, (**c**) trapping experiment of active species during the TC photodegradation, (**d**) six recycling GCN-500 experiments for TC photodegradation, and (**e**) The COD removals of TC with all GCN samples.

**Figure 11 nanomaterials-11-02041-f011:**
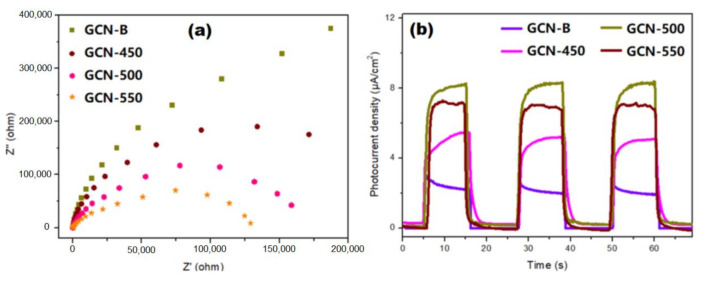
(**a**) EIS changes of GCN-B and GCNs that recalcined at different temperatures, and (**b**) transient photocurrent responses of GCN samples.

**Table 1 nanomaterials-11-02041-t001:** The specific surface area, pore size, and elemental composition of GCN-B and GCNs that recalcined at different temperatures.

Sample	d-Spacing (*Å*)		S_BET_ (m^2^/g)	Pore Size (nm)	Pore Volume(cm^3^/g)	Elemental Composition		C/N Ratio	Yield Remained after Recalcination (%)
	(002)	(100)				C (wt%)	N (wt%)		
GCN-B	3.244	6.783	30.20	3.68	0.083	35.10	51.53	0.681	100
GCN-450	3.220	6.781	79.96	3.49	0.144	35.68	50.33	0.708	73
GCN-500	3.194	6.782	117.05	3.59	0.210	37.71	52.38	0.720	45
GCN-550	3.188	6.781	158.57	3.63	0.228	38.76	47.87	0.809	18

## Data Availability

Not applicable.

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
