# Peer review of "Highly Mesoporous g-C3N4 with Uniform Pore Size Distribution via the Template-Free Method to Enhanced Solar-Driven Tetracycline Degradation"

_nanomaterials, 2021, doi:10.3390/nano11082041_

Round 1

Reviewer 1 Report

The paper presents an attempt to synthetize highly mesoporous graphitic carbon nitrides (GCNs) with uniform pore size distribution of mesopores via template-free method to enhanced solar-driven tetracycline degradation. The presentation of the investigation method as well as scientific results is satisfactory for the paper to be recommended for publication. The minor drawbacks to be addressed can be specified as follows:
1.    Page 1, Abstract. “mesoporous g-C3N4” ---> “mesoporous graphitic carbon nitride g-C3N4”.
2.    Page 1, Keywords. Please add “g-C3N4”.
3.    Page 2, Introduction. Please reject the following sentence “Generally, a porous material with the range of 2-50 nm of pore size is known as mesoporous.” Truism.
4.    Page 2, Introduction. (i) Ge synthesized ---> Ge et al. synthesized. (ii) Hernandez-Uresti produced ---> Hernandez-Uresti et al. produced. Please check all the manuscript.
5.    Pages 2 and 3, Figs. S1 and S2. Please move these figures from Supplementary data to the article!!!! 
6.    Page 3, 2.3. Characterization, “samples were degassed for 4 h at 150 oC”. The typical temperature for such a process is 105-120 oC. Why did the authors use such a high temperature?
7.    Page 4, Tab. 1. “Pore size” ---> “Average pore size”.
8.    Page 5. (i) 30.2 m2/g ---> 30.20 m2/g. See Tab. 1, Pages 7 and 10 (ii) “that is the reason for the low specific surface area (30.2 m2/g)”. The FESEM images cannot be related to the surface area. From these micrograph it is possible discuss roughness. In the other words the visible structure does not necessarily contribute to such a low apparent surface area value at this resolution scale. What is visible is definitely not mesopores!!!
9.    Page 7. “All the pore diameter” ---> “All the pore diameters”.
10.    Page 7, “The result of the BET surface area is well agreed with the FESEM micrograph in Figure 3.”. See point 8.
11.    Page 7. 79% ---> 73%. See Tab. 1.
12.    Page 8, Fig. 6. It is a pity that the authors did not show the N1s results for the remaining samples, for example, in one panel.
13.    Page 10. h+ ---> H+?
14.    Page 11, Fig. 8(b). From the point of view of static analysis the description of the linear equation for GCN-500 data is correct? See the Anscombe's quartet. It is designed for the data with approximately the same linear regression line (as well as nearly identical means, standard deviations, and correlations) but are graphically very different. This illustrates the pitfalls of relying solely on a fitted model to understand the relationship between variables. See (i) https://heap.io/blog/data-stories/anscombes-quartet-and-why-summary-statistics-dont-tell-the-whole-story (ii) https://towardsdatascience.com/importance-of-data-visualization-anscombes-quartet-way-a325148b9fd2 (iii) https://hackernoon.com/anscombes-quartet-and-importance-of-data-visualization-cc163u5r.
15.    Figs. 7 -9. (a), (b), …. Too big letters.
16.    Page 12, Fig. 9(a), Legend. Remove the bold frame.

Author Response

Nanomaterials

Manuscript Draft

Manuscript ID: Nanomaterials-1181641

Reviewer 1:

Comments:

The paper presents an attempt to synthetize highly mesoporous graphitic carbon nitrides (GCNs) with uniform pore size distribution of mesopores via template-free method to enhanced solar-driven tetracycline degradation. The presentation of the investigation method as well as scientific results is satisfactory for the paper to be recommended for publication. The minor drawbacks to be addressed can be specified as follows:

  1. Page 1, Abstract. “mesoporous g-C3N4” ---> “mesoporous graphitic carbon nitride g-C3N4”.

Author response: Thank you for your suggestion. The abstract has been updated.

Abstract: “A highly mesoporous graphitic carbon nitride g-C3N4 (GCN) has been produced…”

  1. Page 1, Keywords. Please add “g-C3N4”.

Author response: Thank you for your suggestion. G-C3N4 has been added as one of the keywords.

“Keywords: Tetracycline; mesoporous, graphitic carbon nitride; uniform pore size distribution; photocatalyst; g-C3N4

  1. Page 2, Introduction. Please reject the following sentence “Generally, a porous material with the range of 2-50 nm of pore size is known as mesoporous.” Truism.

Author response: Thank you for your suggestion. This sentence has been removed.

  1. Page 2, Introduction. (i) Ge synthesized ---> Ge et al. synthesized. (ii) Hernandez-Uresti produced ---> Hernandez-Uresti et al. produced. Please check all the manuscript.

Author response: Thank you for your suggestion. The citation has been revised.

“…Ge et al. synthesized mesoporous GCN for nitrogen photo-fixation application with pore size in the range of 20-160 nm, Hernandez-Uresti et al. produced mesoporous GCN…”

  1. Pages 2 and 3, Figs. S1 and S2. Please move these figures from Supplementary data to the article!!!!

Author response: Thank you for your suggestion, the Fig. S1 and S2 has been moved to article and renamed it as Fig. 1 and Fig. 2.

  1. Page 3, 2.3. Characterization, “samples were degassed for 4 h at 150 oC”. The typical temperature for such a process is 105-120 oC. Why did the authors use such a high temperature?

Author response: Thank you for your question. The degas temperature is according to the standard procedure from the instrument. However, the degas temperature from literature review is varied, for example, Paul et al. conducted GCN degassing at 180 oC [1], Hou et al. degassed GCN composite at 200 oC [2], and Xu et al. degassed GCN composite at 150 oC [3]. Degas temperature need to be below the melting point and/or decomposition temperature. According to Wang et al., the melting point of GCN is 213 oC [4].

  1. Page 4, Tab. 1. “Pore size” ---> “Average pore size”.

Author response: Thank you for your response. The “pore size” has been change to “average pore size”. Besides that, the Table 1 also change to “average pore size”.

“Furthermore, the average pore size of GCN samples was listed in Table 1.”

Table 1. The specific surface area, average pore size, and elemental composition of GCN-B and GCNs that recalcined at different temperature.

Sample

d-spacing (Å)

SBET (m2/g)

Average pore size (nm)

Pore volume

(cm3/g)

Elemental composition

C/N ratio

Yield remained after recalcination (%)

(002)

(100)

C (wt%)

N (wt%)

GCN-B

3.244

6.783

30.20

3.68

0.083

35.10

51.53

0.681

100

GCN-450

3.220

6.781

79.96

3.49

0.144

35.68

50.33

0.708

73

GCN-500

3.194

6.782

117.05

3.59

0.210

37.71

52.38

0.720

45

GCN-550

3.188

6.781

158.57

3.63

0.228

38.76

47.87

0.809

18

  1. Page 5. (i) 30.2 m2/g ---> 30.20 m2/g. See Tab. 1, Pages 7 and 10 (ii) “that is the reason for the low specific surface area (30.2 m2/g)”. The FESEM images cannot be related to the surface area. From these micrograph it is possible discuss roughness. In the other words the visible structure does not necessarily contribute to such a low apparent surface area value at this resolution scale. What is visible is definitely not mesopores!!!

Author response: Thank you for your comment. The sentence has been removed, and the paragraph has been revised.

Result & discussion: Fig. 5 displays the morphology of GCN samples. The GCN-B shows disordered bulks stacks together. Obviously, GCN-B has a typical non-porous architecture and slightly flat. Fig. 5(b) shows a flake-like surface of GCN-450, some of the holes were exposed on the surface. Meanwhile, refer to Fig. 5(c), the surface of GCN-500 distributed with numerous inhomogeneous holes and uneven surface. This finding was concordance with Papailias et al. [5] because when the temperature of recalcination raise to 550 oC, the GCN structure shows loose surface morphologies. Furthermore, this experiment did not report the 600 oC of GCN recalcination because it was found that the GCN was fully disappeared after recalcination.

  1. Page 7. “All the pore diameter” ---> “All the pore diameters”.

Author response: Thank you for your response. The “all the pore diameter” has been change to “all the pore diameters”

Page 7: “All the pore diameters of GCNs…”

  1. Page 7, “The result of the BET surface area is well agreed with the FESEM micrograph in Figure 3.”. See point 8.

Author response: Thank you for your response. This sentence has been removed.

  1. Page 7. 79% ---> 73%. See Tab. 1.

Author response: Thank you for your comment. The 79% in manuscript has been changed to 73%.

Result & discussion: “The yield of GCN-450 and GCN-500 is 73% and 45%...”

  1. Page 8, Fig. 6. It is a pity that the authors did not show the N1s results for the remaining samples, for example, in one panel.

Author response: Thank you for your suggestion. The N1s of GCN-450 and GCN-550 has been added in supplementary file.

Fig. S3. The XPS of GCN-450 and GCN-550.

  1. Page 10. h+ ---> H+?

Author response: Thank you for your comment. The h+ has been change to H+.

Page 10:                                                                      (Eq. 3)

  1. Page 11, Fig. 8(b). From the point of view of static analysis the description of the linear equation for GCN-500 data is correct? See the Anscombe's quartet. It is designed for the data with approximately the same linear regression line (as well as nearly identical means, standard deviations, and correlations) but are graphically very different. This illustrates the pitfalls of relying solely on a fitted model to understand the relationship between variables. See (i) https://heap.io/blog/data-stories/anscombes-quartet-and-why-summary-statistics-dont-tell-the-whole-story (ii) https://towardsdatascience.com/importance-of-data-visualization-anscombes-quartet-way-a325148b9fd2 (iii) https://hackernoon.com/anscombes-quartet-and-importance-of-data-visualization-cc163u5r.

Author response: Thank you for your comment. The data was calculated by using two models, which are kinetic 1st order and kinetic 2nd order [6], it was found that the kinetic 1st order is more fitted to my outcome.  

Fig. S2. Fitting curve of GCNs using different kinetics reaction equation.

  1. Figs. 7 -9. (a), (b), …. Too big letters.

Author response: Thank you for your suggestions. The alphabet in the Figures has been edited to smaller.

Fig. 9. (a) UV-Vis absorption spectrum; (b) plots of (αhv)1/2 vs. photon energy of the GCN-B and GCN that recalcined at different temperature; (c) PL spectra of GCN-B and GCN that recalcined at different temperature.

Fig. 10. (a) Photocatalytic performance of TC with GCN samples under irradiation of visible light (λ > 420 nm), (b) the kinetics for TC photodegradation with different GCN samples, (c) trapping experiment of active species during the TC photodegradation, (d) Six recycling GCN-500 experiments for TC photodegradation, and (e) The COD removals of TC with all GCN samples.

Fig. 11(a). EIS changes of GCN-B and GCNs that recalcined at different temperature, and (b) Transient photocurrent responses of GCN samples.

  1. Page 12, Fig. 9(a), Legend. Remove the bold frame.

Author response: Thank you for your suggestion. The bold frame has been removed.

Fig. 11(a). EIS changes of GCN-B and GCNs that recalcined at different temperature, and (b) Transient photocurrent responses of GCN samples.

  1. Paul, D.R., et al., Effect of calcination temperature, pH and catalyst loading on photodegradation efficiency of urea derived graphitic carbon nitride towards methylene blue dye solution. RSC advances, 2019. 9(27): p. 15381-15391.
  2. Hou, L.-p., et al., Preparation of MoO2/g-C3N4 composites with a high surface area and its application in deep desulfurization from model oil. Applied Surface Science, 2018. 434: p. 1200-1209.
  3. Xu, Q., et al., Making co-condensed amorphous carbon/g-C3N4 composites with improved visible-light photocatalytic H2-production performance using Pt as cocatalyst. Carbon, 2017. 118: p. 241-249.
  4. Wang, X.L., et al., Bottom-Up Enhancement of g-C3N4 Photocatalytic H2 Evolution Utilising Disordering Intermolecular Interactions of Precursor. International Journal of Photoenergy, 2014. 2014.
  5. Papailias, I., et al., Chemical vs thermal exfoliation of g-C3N4 for NOx removal under visible light irradiation. Applied Catalysis B: Environmental, 2018. 239: p. 16-26.
  6. Gao, W., et al., The role of reduction extent of graphene oxide in the photocatalytic performance of Ag/AgX (X= Cl, Br)/rGO composites and the pseudo-second-order kinetics reaction nature of the Ag/AgBr system. Physical Chemistry Chemical Physics, 2016. 18(27): p. 18219-18226.
  7. Fu, L., X. Xiao, and A. Wang, Reduced graphene oxide coupled with g-C3N4 nanodots as 2D/0D nanocomposites for enhanced photocatalytic activity. Journal of Physics and Chemistry of Solids, 2018. 122: p. 104-108.
  8. Sudrajat, H. and S. Hartuti, One-pot, solid-state loading of Zn into g-C3N4 for increasing the population of photoexcited electrons and the rate of photocatalytic hydrogen evolution. Optik, 2019. 181: p. 1057-1065.
  9. Zhou, X., et al., Three dimensional hierarchical heterostructures of g-C3N4 nanosheets/TiO2 nanofibers: controllable growth via gas-solid reaction and enhanced photocatalytic activity under visible light. Journal of hazardous materials, 2018. 344: p. 113-122.
  10. Liu, X., et al., The synergy of thermal exfoliation and phosphorus doping in g-C3N4 for improved photocatalytic H2 generation. International Journal of Hydrogen Energy, 2021. 46(5): p. 3595-3604.
  11. Wang, H., et al., Enhanced photocatalytic hydrogen production of restructured B/F codoped g-C3N4 via post-thermal treatment. Materials Letters, 2018. 212: p. 319-322.
  12. Ong, W.-J., et al., Graphitic carbon nitride (g-C3N4)-based photocatalysts for artificial photosynthesis and environmental remediation: are we a step closer to achieving sustainability? Chemical reviews, 2016. 116(12): p. 7159-7329.
  13. Qin, J., et al., Two-dimensional porous sheet-like carbon-doped ZnO/g-C3N4 nanocomposite with high visible-light photocatalytic performance. Materials Letters, 2017. 189: p. 156-159.
  14. Dong, F., et al., Enhanced visible light photocatalytic activity and oxidation ability of porous graphene-like g-C3N4 nanosheets via thermal exfoliation. Applied Surface Science, 2015. 358: p. 393-403.
  15. Li, Y., et al., Cross‐Linked g‐C3N4/rGO Nanocomposites with Tunable Band Structure and Enhanced Visible Light Photocatalytic Activity. Small, 2013. 9(19): p. 3336-3344.

Reviewer 2 Report

Comments and suggestions to the authors:

  1. The bandgap reported in the literature is 2.7eV, while in the article a value of 2.68eV was referred as improved. Is the difference of 0.02eV is really significant?
  2. In the introduction, at the third paragraph, it is written that the bandgap is 2.8eV which is not correct.
  3. The preparation of mesoporous GCN (recalcination) is not clear: what was the heating rate? Did the temperature stay a certain time at maximum? If so, for how long?
  4. I may not understand this correctly, but how did the authors obtain TEM images of the crystalline structure? Isn’t it something you obtain for metals? Other than that, they claim that additional calcination creates thinner and scattered layers. Shouldn’t it "break" the crystalline structure (if it indeed exists)?
  5. On page 7 they indicate that a higher temp of recalcination causes the material to be more porous as there is a more massive release of gases. It is not clear. What happens in the recalcination that does not occur in the first calcination? Since the initial heating takes place already at a high temperature, up to 550 Celsius, all the gases that were supposed to be released have already been released (water, ammonia). In my opinion this assumption is wrong and it is simply a matter of disassembly of the material (The authors also state this fact at the beginning of page 8).
  6. Following Note 5, I think it is worthwhile to perform a TGA measurement to examine the thermal stability of the GCN they fabricated. My assumption is that their material will break down well before 600 degrees (as usually happens).
  7. Because the authors indicate that it is also a breakdown of the substance, which causes the formation of pores - Is the structure of GCN not broken? Maybe another material was created, with the same composition, but with greater porosity?
  8. On page 8, XPS analysis, there is no preservation of the electronic structure, while the authors claim it is so because the material is oxidized. I think they should provide the peaks for O1s to see if there is a change.
  9. On page 9, the bandgap data is not consistent. In the introduction the writers state a value of 2.8eV. In measurements for GCN-B which should be compatible with the literature they received 2.62eV. In the literature the value is 2.7eV.

I think these measurements need to be repeated.

  1. Following comment 9, the explanation for the reason why GCN-500 is better photocatalytically than GCN-550 is incorrect.

Author Response

Nanomaterials

Manuscript Draft

Manuscript ID: Nanomaterials-1181641

Reviewer 2:

 Comments:

  1. The bandgap reported in the literature is 2.7eV, while in the article a value of 2.68eV was referred as improved. Is the difference of 0.02eV is really significant?

Author response: Thank you for your comment. According to literature review, some researchers synthesized GCN which is higher than 2.7 eV. For example, Fu et al. synthesized GCN with 2.8 eV [7] and Sudrajat et al. synthesized pristine GCN with 2.8 eV [8]. Meanwhile, Zhou et al. synthesized pure GCN with 2.9 eV [9]. As a result, the band gap of GCN is in between 2.4-2.9 eV, it is depending on the preparation method. In this study, the best obtain bandgap value for GCN is 2.62 eV, which is bulk GCN, and the highest performance was GCN-500, 2.65 eV (after repeat the UV-DRS). However, the bandgap of GCN will be slightly blue shift based on recalcination. This finding is similar to Liu et al. but this author was conducted in different recalcination condition [10].

  1. In the introduction, at the third paragraph, it is written that the bandgap is 2.8eV which is not correct.

Author response: Thank you for your comment. The bandgap value has been changed as response in comment 1.

Introduction: “…due to visible light active with a band gap of 2.4-2.9 eV..”

  1. The preparation of mesoporous GCN (recalcination) is not clear: what was the heating rate? Did the temperature stay a certain time at maximum? If so, for how long?

Author response: Thank you for your comment. The heating rate and calcination time has been added in section 2.1 and 2.2.

Section 2.1: “It was calcined at 550 oC in air for 4 h at 3.0 oC min-1 heating rate as shown…”

Section 2.2: “…at three different temperatures (450, 500, and 550 oC) for 2 h at 3.0 oC min-1 to…”

  1. I may not understand this correctly, but how did the authors obtain TEM images of the crystalline structure? Isn’t it something you obtain for metals? Other than that, they claim that additional calcination creates thinner and scattered layers. Shouldn’t it "break" the crystalline structure (if it indeed exists)?

Author response: Thank you for your comment. The TEM images consists of amorphous and crystalline structure of GCN. When further enlarge the crystalline part, we can find out the lattice d-spacing image. Meanwhile, the GCN is 2D layer structure, like graphene, it is a sheet-like structure. Under high temperature recalcination, the GCN layer become smaller than GCN-B due to mass loss during recalcination and the large layers are split into small nanosheets during recalcination. As a result, recalcination is a process that could enlarge the specific area and create porous in GCN by splitting the layers and releasing gaseous products. Furthermore, the GCN structures has been confirm by using XRD, which is well-fitted with JCPDS 87-1526.

  1. On page 7 they indicate that a higher temp of recalcination causes the material to be more porous as there is a more massive release of gases. It is not clear. What happens in the recalcination that does not occur in the first calcination? Since the initial heating takes place already at a high temperature, up to 550 Celsius, all the gases that were supposed to be released have already been released (water, ammonia).

In my opinion this assumption is wrong and it is simply a matter of disassembly of the material (The authors also state this fact at the beginning of page 8).

Author response: Thank you for your response. In the 1st calcination, it is formation of GCN from melamine (precursor). The formation of GCN is a process of thermal polymerization, where melamine decomposes by releasing ammonia gas to form melam, and releasing another ammonia to form melem as shown in figure below. This melem will undergo further thermal polymerization to form melon and finally polymeric GCN is formed.

For the 2nd calcination, the GCN has already form, however, the structure of GCN might be bulk and low surface area. Therefore, the 2nd calcination is to further decompose part of the GCN and creates numerous pores and holes in the bulk GCN to make it as loose structure. In 2018, Wang et al. reported post-thermal treatment of doped GCN with similar finding [11].

  1. Following Note 5, I think it is worthwhile to perform a TGA measurement to examine the thermal stability of the GCN they fabricated. My assumption is that their material will break down well before 600 degrees (as usually happens).

Author response: Thank you for your comment. Figure below shows the TGA of GCN and it was added in supplementary file. It was found that starting from 500 oC, GCN will slowly decompose. Ong et al. reported that GCN can decompose fully at 600-650 oC [12]. In this study, it was found that the mass of GCN after recalcination was getting lower, this is due to GCN was in the process of breaking down. As mentioned in this study, after 2 h of recalcination at 550 oC, the GCN was remain 18%. If the recalcination temperature increase to 600 oC, GCN was fully decomposed, that is the reason why this work does not conduct recalcination at 600 oC.

Figure S1. The TGA of GCN-B.

  1. Because the authors indicate that it is also a breakdown of the substance, which causes the formation of pores - Is the structure of GCN not broken? Maybe another material was created, with the same composition, but with greater porosity?

Author response: Thank you for your question. There are 2 kinds of GCNs available, which are, triazine and heptazine. According to Liu et al., triazine based GCN will show 6 peaks in the XRD, which are (100), (110), (200), (002), (102), and (210) planes. In this study, the XRD only shows 2 peaks, which are (100) and (002), indicate that heptazine GCN is formed in this study. FTIR shows that GCN at 3000-3400 cm-1 has a weak broad peak, usually, triazine GCN will have stronger broad band at the range due to high amount of -NH groups. In addition, the ratio of NC3 to C-N=C is this study is 1:5.5, which is close to the theoretical value (1:6) of the heptazine GCN. As a result, this work confirmed that heptazine GCN is produced.

  1. On page 8, XPS analysis, there is no preservation of the electronic structure, while the authors claim it is so because the material is oxidized. I think they should provide the peaks for O1s to see if there is a change.

Author response: Thank you for your suggestion. The O1s of GCN-B and GCN-500 have similar peak, which are 531.8 eV represents to C-O bond, 533.3 eV and 534.9 eV corresponding to surface -OH and chemisorbed H2O.

Fig. S4. The O1s peak of GCN-B and GCN-500.

  1. On page 9, the bandgap data is not consistent. In the introduction the writers state a value of 2.8eV. In measurements for GCN-B which should be compatible with the literature they received 2.62eV. In the literature the value is 2.7eV.

I think these measurements need to be repeated.

Author response: Thank you for your comment. The absorption test has been repeated. It was found that the band gap of GCNs is similar. The band gap of GCN that found in literature is not only 2.7 eV. However, the band gap value of GCN can be in between 2.4-2.9 eV. For example, Qin et al. synthesized pure GCN with only 2.4 eV [13], Dong et al. synthesized bulk GCN with 2.42 eV, [14], Li et al. synthesized pure GCN with 2.5 eV [15]. As a result, the band gap obtained in this study is within the range of the literature reported GCN band gap (2.4-2.9 eV).

  1. Following comment 9, the explanation for the reason why GCN-500 is better photocatalytically than GCN-550 is incorrect.

Author response: Thank you for your comment. The paragraph has been revised.

Result & Discussion: The adsorption capacity gradually increases with the recalcination temperature. Fig. 10(a) shows the photodegradation performance of 10 mg/L TC in visible light irradiation. It was found that recalcined GCN has better dark adsorption than GCN-B, and the adsorption capacity was gradually increase with the increase of recalcination temperature. It was noticed that the photodegradation efficiency sequences as follows GCN-500 (0.0247 min-1, 99.4%) > GCN-550 (0.0158 min-1, 96.0%) > GCN-450 (0.0137 min-1, 93.5%) > GCN-B (0.0116 min-1, 87.8%). The GCN-500 show highest photodegradation activity because of the large specific surface area (117.05 m2/g). Large specific surface area indicates that more active sites on GCN-500, that able to receive light irradiation and enhanced that photocatalysis process, confirming that 500 oC is the optimal recalcination temperature. In addition, GCN-500 has visible light band gap, which is 2.65 eV that corresponding to 468 nm. With these two criteria, visible light band gap and large surface area and pore volume, the photoactivity of GCN-500 to degrade TC has greatly improved. This is because GCN-500 are highly mesoporous that is rich in active sites.

  1. Paul, D.R., et al., Effect of calcination temperature, pH and catalyst loading on photodegradation efficiency of urea derived graphitic carbon nitride towards methylene blue dye solution. RSC advances, 2019. 9(27): p. 15381-15391.
  2. Hou, L.-p., et al., Preparation of MoO2/g-C3N4 composites with a high surface area and its application in deep desulfurization from model oil. Applied Surface Science, 2018. 434: p. 1200-1209.
  3. Xu, Q., et al., Making co-condensed amorphous carbon/g-C3N4 composites with improved visible-light photocatalytic H2-production performance using Pt as cocatalyst. Carbon, 2017. 118: p. 241-249.
  4. Wang, X.L., et al., Bottom-Up Enhancement of g-C3N4 Photocatalytic H2 Evolution Utilising Disordering Intermolecular Interactions of Precursor. International Journal of Photoenergy, 2014. 2014.
  5. Papailias, I., et al., Chemical vs thermal exfoliation of g-C3N4 for NOx removal under visible light irradiation. Applied Catalysis B: Environmental, 2018. 239: p. 16-26.
  6. Gao, W., et al., The role of reduction extent of graphene oxide in the photocatalytic performance of Ag/AgX (X= Cl, Br)/rGO composites and the pseudo-second-order kinetics reaction nature of the Ag/AgBr system. Physical Chemistry Chemical Physics, 2016. 18(27): p. 18219-18226.
  7. Fu, L., X. Xiao, and A. Wang, Reduced graphene oxide coupled with g-C3N4 nanodots as 2D/0D nanocomposites for enhanced photocatalytic activity. Journal of Physics and Chemistry of Solids, 2018. 122: p. 104-108.
  8. Sudrajat, H. and S. Hartuti, One-pot, solid-state loading of Zn into g-C3N4 for increasing the population of photoexcited electrons and the rate of photocatalytic hydrogen evolution. Optik, 2019. 181: p. 1057-1065.
  9. Zhou, X., et al., Three dimensional hierarchical heterostructures of g-C3N4 nanosheets/TiO2 nanofibers: controllable growth via gas-solid reaction and enhanced photocatalytic activity under visible light. Journal of hazardous materials, 2018. 344: p. 113-122.
  10. Liu, X., et al., The synergy of thermal exfoliation and phosphorus doping in g-C3N4 for improved photocatalytic H2 generation. International Journal of Hydrogen Energy, 2021. 46(5): p. 3595-3604.
  11. Wang, H., et al., Enhanced photocatalytic hydrogen production of restructured B/F codoped g-C3N4 via post-thermal treatment. Materials Letters, 2018. 212: p. 319-322.
  12. Ong, W.-J., et al., Graphitic carbon nitride (g-C3N4)-based photocatalysts for artificial photosynthesis and environmental remediation: are we a step closer to achieving sustainability? Chemical reviews, 2016. 116(12): p. 7159-7329.
  13. Qin, J., et al., Two-dimensional porous sheet-like carbon-doped ZnO/g-C3N4 nanocomposite with high visible-light photocatalytic performance. Materials Letters, 2017. 189: p. 156-159.
  14. Dong, F., et al., Enhanced visible light photocatalytic activity and oxidation ability of porous graphene-like g-C3N4 nanosheets via thermal exfoliation. Applied Surface Science, 2015. 358: p. 393-403.
  15. Li, Y., et al., Cross‐Linked g‐C3N4/rGO Nanocomposites with Tunable Band Structure and Enhanced Visible Light Photocatalytic Activity. Small, 2013. 9(19): p. 3336-3344.
